# Characterization of a New Citrus Mutant Induced by Gamma Irradiation with a Unique Fruit Shape, Gwonje-Early, and Determination of Specific Selection Markers Using Allele-Specific PCR

**DOI:** 10.3390/plants13060911

**Published:** 2024-03-21

**Authors:** Chang-Ho Eun, Jung-Gwon Ko, In-Jung Kim

**Affiliations:** 1Subtropical Horticulture Research Institute, Jeju National University, Jeju-Si 63243, Republic of Korea; 2Faculty of Biotechnology, College of Applied Life Sciences, Jeju National University, Jeju-Si 63243, Republic of Korea; 3Research Institute for Subtropical Agriculture and Biotechnology, SARI, Jeju National University, Jeju-Si 63243, Republic of Korea

**Keywords:** *Citrus unshiu*, gamma ray, SNP, InDel, allele-specific PCR

## Abstract

Gamma-ray irradiation is one of the most widely used mutagens worldwide. We previously conducted mutation breeding using gamma irradiation to develop new *Citrus unshiu* varieties. Among these mutants, Gwonje-early had an ovate shape, a protrusion of the upper part of the fruit, and a large fruit size compared with wild-type (WT) fruits. We investigated the external/internal morphological characteristics and fruit sugar/acid content of Gwonje-early. Additionally, we investigated genome-wide single-nucleotide polymorphisms (SNPs) and insertion/deletion (InDel) variants in Gwonje-early using whole-genome re-sequencing. Functional annotation by Gene Ontology analysis confirmed that InDels were more commonly annotated than SNPs. To identify specific molecular markers for Gwonje-early, allele-specific PCR was performed using homozygous SNPs detected via Gwonje-early genome re-sequencing. The GJ-SNP1 and GJ-SNP4 primer sets were effectively able to distinguish Gwonje-early from the WT and other commercial citrus varieties, demonstrating their use as specific molecular markers for Gwonje-early. These findings also have important implications in terms of intellectual property rights and the variety protection of Gwonje-early. Our results may provide insights into the understanding of morphological traits and the molecular breeding mechanisms of citrus species.

## 1. Introduction

Citrus fruits are some of the most economically important fruits worldwide, containing high levels of vitamin C and macro- and micronutrients [1]. Miyagawa-wase (*Citrus unshiu* Marcow), an early-season citrus variety, is one of the most widely cultivated varieties, accounting for more than 80% of the total citrus fruit cultivation on Jeju Island in Korea. According to a 2023 report from the National Statistical Office, the citrus fruit cultivation area on Jeju Island was the largest cultivation area at 15,818 ha, followed by that of plums and persimmons, accounting for 56 ha and 55 ha, respectively. The improvement of citrus cultivars through traditional breeding is difficult because of several reproductive biological characteristics, including apomixis, the partial sterility of both sexes, mating incompatibility, a long juvenile period, and high heterozygosity [2,3,4]. Transgenic plant approaches introducing genes to produce good traits can also be used to develop superior citrus varieties, but restrictions on genetically modified organisms make them difficult to sell on global markets [5].

Mutation is ‘the process by which genes are permanently altered under environmental conditions while being transferred between generations’ [6]. Mutation breeding is an alternative method for breeders because it provides an opportunity to obtain desired traits that do not exist in nature or that have been lost during evolution and thus greatly contributes to the creation of new plant varieties and genetic resources. Gamma-ray irradiation is the most widely used mutagen worldwide according to the Joint Food and Agriculture Organization/International Atomic Energy Agency Mutant Variety Database (Joint FAO/IAEA). Cobalt-60 and cesium-137 are the main sources of gamma rays for radiobiological work [7,8,9]. Ease of use has played an important role in the spread of applications using gamma rays for the development of new crop varieties. New species of many plants have been obtained, such as coriander [10], tomato [11], Anthurium [12], and mung bean [13], using gamma irradiation. Mutant breeding research using gamma rays has a long history but has not been continuously reported until recently. The semi-dwarf wheat mutant *jg0030* was identified by the gamma-ray mutagenesis of the wheat variety ‘Jing411’ (wild type, WT) and showed no yield penalty [14]. Compared with ‘Jing411’, the plant height of the *jg0030* mutant was reduced by 7–18% over a 2-year field experiment, and no changes were observed in the yield-related traits of the plants. Abe et al. (2023) identified a pollen-part self-compatible apple (*Malus* × *domestica* Borkh.) mutant, Morioka #61-G-827, induced by gamma-ray mutagenesis [15]. In addition to this study, our mutant citrus breeding research also recently reported a new variety developed using gamma rays. The *C. unshiu* mutant *Jedae-unshiu* has a fruit peel with a vertical groove and a smooth albedo, with good adhesion between the peel and flesh. Additionally, it had thicker skin and greater fruit hardness than the WT control. The contents of the representative flavonoids hesperetin and narirutin were also higher in the skin and flesh of *Jedae-unshiu* than in the WT control [16]. We also reported another new *C. unshiu* mutant, Ara-unshiu, with a unique, late-fruit-ripening phenotype [17].

Fruit size has become as important as quantity in determining the profitability of citrus cultivation, and an economic premium is generally obtained by increasing fruit size instead of increasing crop yield. Several techniques, such as hand thinning; chemical and hormonal thinning; and hormonal stimulation to speed up fruit growth, have been attempted to increase fruit size beyond the limits achievable through the optimization of standard growing practices (fertilization, irrigation, and pruning) [18,19,20,21,22,23]. In the present study, we uncovered a novel *C. unshiu* mutant with altered fruit size and shape, Gwonje-early, which exhibits an ovate shape, protrusion of the upper part of the fruit, and a large fruit size compared with WT fruit. In addition, we conducted whole-genome re-sequencing to identify specific molecular markers for the Gwonje-early mutant. Our results contribute to the understanding of the processes and the molecular mechanisms involved in fruit size and shape.

## 2. Results

### 2.1. Selection of the Gwonje-Early Mutant Line by Gamma Irradiation and Its Morphological Traits

After irradiating gamma rays (60 Co; 100 Gy; shoot survival rate, 60%) were applied to the scion branches of *C. unshiu* (Marc. cv. Miyagawa-wase), they were grafted onto the branches of Miyagawa-wase (*C. unshiu* Marc) [24]. Mutant lines were selected by investigating the fruits from branches grafted that year. Among the many mutant lines generated in this study, we selected a mutant line named Gwonje-early, which exhibited a variation in fruit appearance compared with WT fruit. The following year, only Gwonje-early scions were grafted onto rootstock (*C. unshiu* Marc. cv. Miyagawa-wase) and were continuously monitored to ensure that their characteristics were maintained [24,25].

### 2.2. Morphological Traits of Gwonje-Early and WT Fruits

The external morphological differences between Gwonje-early and WT fruit were investigated. At the mature fruit stage (late November of 2022 and 2023), Gwonje-early exhibited an ovate shape, with the upper part of the fruit protruding and a large fruit size compared with WT fruit (Figure 1). In a comparative analysis of other morphological and physiological traits between Gwonje-early and WT fruits produced in 2022 and 2023, Gwonje-early fruits showed greater horizontal (68.24 ± 3.81 mm in 2022 and 65.70 ± 3.66 mm in 2023) and vertical length (78.63 ± 4.55 mm in 2022 and 84.93 ± 4.68 mm in 2023), fruit weight (195.7 ± 21.60 g in 2022 and 217.6 ± 21.40 g in 2023), peel thickness (3.06 ± 0.64 mm in 2022 and 3.60 ± 0.43 mm in 2023), and fruit hardness (1071 ± 159 G in 2022 and 1196 ± 476 G in 2023) than the WT fruits (Table 1). A comparison of Hunter color values between Gwonje-early and WT fruit peels showed similarities in the lightness value, red value, and yellow value. The sugar levels and acidity were also similar between Gwonje-early and WT fruits (Table 1).

### 2.3. Mapping of Re-Sequencing Reads to C. unshiu Marc. Miyagawa-Wase CUMW_v1.0

We recently reported that two citrus mutants, Jedae-unshiu and Ara-unshiu, identified by gamma irradiation showed a unique fruit phenotype, with vertical troughs on the flavedo and late ripening, respectively [16,17]. To better understand the phenotypes of Gwonje-early mutant fruits, such as the upper protuberance, large fruit, thick pericarp, and increased hardness, we first performed whole-genome re-sequencing of Gwonje-early mutant fruit to identify genome sequence polymorphisms between Gwonje-early and WT fruits. The genome sequence data of WT fruit used in this study were from our previously published sources (NCBI: PRJNA745525) [26]. *C. unshiu* Marc. Miyagawa-wase (CUMW_v1.0) was used as a reference genome and had a length of 359.7 Mb [27]. After sequence pre-processing the raw data from Gwonje-early and the WT control, we obtained 96,356,847 and 80,693,250 clean reads, respectively. The mapped region rates were 86.88% and 86.75%, and the average coverage per sample was 29.97× and 27.49× for Gwonje-early and the WT, respectively (Table 2).

Compared with the reference genome, 1,198,650 and 1,204,414 SNPs were identified in WT and Gwonje-early plants, respectively (Appendix A). Of these, there were 8208 and 7339 homozygous SNPs and 572,811 and 604,548 heterozygous SNPs in the WT and Gwonje-early plants, respectively. These SNPs were classified by gene annotation (Appendix A). In the WT, 367,591 SNPs were detected in genic regions (164,743 in exons and 212,665 in introns), and 715,536 SNPs were detected in intergenic regions. In Gwonje-early, 370,813 SNPs were detected in genic regions (165,449 in exons and 215,387 in introns), and 719,688 SNPs were detected in intergenic regions. We also detected InDels in the WT and Gwonje-early genome sequences after comparison with the reference genome (Appendix A). In the WT plants, 172,259 total InDels were detected, of which 3751 (1600 insertions and 2151 deletions) and 57,930 (29,138 insertions and 28,794 deletions) homozygous and heterozygous InDels were found, respectively. In Gwonje-early, 167,284 total InDels were detected, of which 4.050 (1693 insertions and 2357 deletions) and 55,773 (27,804 insertions and 27,969 deletions) homozygous and heterozygous InDels were found, respectively. These InDels were classified by gene annotation (Appendix A). In the WT plants, 45,362 InDels were detected in genic regions (11,128 in exons and 35,441 in introns), and 110,566 InDels were detected in intergenic regions. In Gwonje-early, 44,352 InDels were detected in genic regions (10,930 in exons and 34,601 in introns), and 100,902 InDels were detected in intergenic regions.

### 2.4. Genetic Variation between WT and Gwonje-Early Plants Based on SNPs and InDels

To identify the mutation(s) responsible for the protrusion of the upper part of the fruit and the large size, thicker peel, and greater hardness of Gwonje-early, we compared the genomic sequences of WT and Gwonje-early plants and uncovered 6132 SNPs (37 homozygous and 6096 heterozygous) and 5969 InDels (29 homozygous and 5940 heterozygous) (Table 3). Of these, we further analyzed SNP and InDel variation in the gene region. In total, 3344 SNPs and 3154 InDels and 465 SNPs and 709 InDels were detected in the genes annotated in the GO database from WT and Gwonje-early plants, respectively (Table 4). When we further filtered for homo-type SNPs and InDels, two SNPs and one InDel were found in the gene regions. The two SNPs were annotated in the glutamate receptor 3.2 gene of *C. sinnensis* and the hypothetical protein CUMW_259270 of *C. unshiu*, and the one InDel was annotated in NO-associated protein 1, a chloroplastic/mitochondrial isoform X1 gene of *C. clementina* (Appendix A).

### 2.5. Functional Annotation of SNP and InDel Gene Variants of Ara-Unshiu

Non-synonymous SNPs and InDels that occur in the coding sequence (CDS) region could have altered gene functions. Thus, we identified SNP and InDel variants of the functional genes of Gwonje-early annotated in the GO database. The identified variants were classified into biological process (BP), cellular component (CC), and molecular function (MF) categories. (Figure 2). In the GO analysis of the InDel variants, BP genes (40,473) were classified into 2003 subcategories, with biological process (963), cellular process (693), metabolic process (593), organic substance metabolic process (562), and primary metabolic process (517) accounting for the largest proportion. CC genes (10,711) were classified into 259 subcategories, including cellular component (992), cellular anatomical entity (958), intracellular anatomical structure (737), intracellular organelle (631), and organelle (631). MF genes (10,748) were classified into 632 subcategories, including molecular function (952), binding (546), catalytic activity (472), organic cyclic compound binding (347), and ion binding (281). (Figure 2A). In the GO analysis of SNP variants, BP genes (72,394) were classified into 2268 subcategories, with biological process (1648), cellular process (1232), metabolic process (1089), organic substance metabolic process (1028), and primary metabolic process (974) accounting for the largest proportion. CC genes (19,095) were classified into 370 subcategories, including cellular component (1719), cellular anatomical entity (1686), intracellular anatomical structure (1362), organelle (1193), and intracellular organelle (1191). MF genes (17,172) were divided into 891 subcategories, including molecular function (1668), binding (910), catalytic activity (866), organic cyclic compound binding (511), and ion binding (482) (Figure 2B). These results showed that the InDel variants had more annotations in the GO database than the SNP variants did, and the top five BP, CC, and MF subcategories were identically annotated between SNP and InDel variants. 

### 2.6. Identification of an Allele-Specific Marker for Ara-Unshiu

AS-PCR is a method used to detect SNPs based on the extension of PCR primers only when the 3′ end of the PCR primer is perfectly complementary to the template [28,29]. To identify mutation-specific selection markers for Gwonje-early using AS-PCR technology, five homo-type SNPs (GJ-SNP1 to 5) were selected from the SNP variants between Gwonje-early and the WT plants. Of the selected five GJ-SNPs, GJ-SNP1 was in the 5′-UTR of the glutamate receptor 3.2 gene, and the others were in intergenic regions. Then, a pair of common primers and a specific primer with an introduced mismatch in the vicinity of the SNP site were designed (Appendix A). Of the five sets of primers, two primer sets, GJ-SNP1 and GJ-SNP4, were effectively able to distinguish Gwonje-early from the WT and other citrus varieties (CG, GP, HB, and KH) (Figure 3).

## 3. Discussion

Fruit shape and size are ultimately determined by the coordinated progression of cell production and cell expansion during fruit growth and development [30]. Therefore, genes that regulate cell production and/or cell expansion may regulate the final fruit size. Several mutations in quantitative trait loci that affect fruit size and/or shape have been identified in tomatoes, apples, and cucumbers. A microtubule-associated protein, SlMAP70, identified from tomatoes functions as an important regulator of fruit shape [31]. Transgenic tomatoes overexpressing the SlMAP70-1 gene show an elongated fruit phenotype due to reduced cell circularity and microtubule anisotropy, whereas loss-of-function mutant fruits show the opposite phenotype, with a flatter shape than the control. A strong ABA-deficient *notabilis*/*flacca* double mutant from tomatoes was also found to be related to fruit size [32]. These mutant lines show that reduced fruit size is caused by an overall smaller cell size. In apples, a spontaneous mutant of the cultivar Gala, Grand Gala, with a large fruit size was identified, which was 15% larger in diameter and 38% heavier than Gala fruit [33]. This result was largely due to altered cell production and enhanced cell size. In cucumbers, the *CsTRM5* gene regulates fruit shape by influencing the cell division direction and cell expansion, and ABA participates in *CsTRM5*-mediated cell expansion during fruit elongation [34]. The tomato *OVATE* gene regulates the gynoecium shape by controlling the cell division pattern in both the proximal–distal and medial–lateral directions [35]. Another tomato gene, *SUN*, which encodes a calmodulin-binding protein, functions in specifying fruit shape by increasing cell division in the longitudinal direction and decreasing cell division in the transverse direction of the fruit [36].

The statistical pattern of SNP and InDel detection and classification via genome annotation using the reference genome (*C. unshiu* Marc. Miyagawa-wase (CUMW_v1.0)) showed a similar distribution between Gwonje-early and WT plants (Appendix A). In both Gwonje-early and WT plants, the SNPs and InDels tended to be mainly distributed in intergenic rather than genic regions. In the comparison between the WT and reference genomes, many SNPs and InDels were detected in the WT despite being the same cultivar, suggesting that the published reference genome sequence is significantly different from our WT sequence. We previously reported that *C. unshiu* Marc. cv. Miyagawa-wase cultivated in a different region shows genome-wide DNA variations including SNPs and InDels, suggesting that the genome re-sequencing of control samples should be considered in genome-wide comparative studies. Torkamaneh et al. (2018) reported that the sequence alignments of reference genomes from two different geographical regions showed the presence of structural variants in individual samples [37]. In comparisons of genetic variation between Gwonje-early and WT plants, we could reduce many of the non-specific SNPs and InDels generated when compared with the reference *C. unshiu* genome, showing that the specific SNPs and/or InDels from Gwonje-early can be extracted by comparing its WT genome sequences. These results also indicate that Gwonje-early genome sequences have more InDel polymorphism mutations than SNP mutations. The frequencies of polymorphic SNPs and InDels were also similar to those found in our previous studies using the genome re-sequencing of Jedae-unshiu and Ara-unshiu mutants [16,17].

AS-PCR is simple, rapid, economical, and reliable and does not require high-level skills and equipment. Therefore, it has been widely used to screen genetic resources for target SNP genotypes in many natural populations or mutant segregation populations [38,39,40,41]. The location of the introduced mismatched bases is critical to the success of AS-PCR. The penultimate (−2 or −3) position of the 3′ end of the mismatch primer is typically used to introduce mismatch bases to maximize detection specificity and prevent false positives [42,43]. In the present study, a forward primer for an SNP with a mismatch in the second position was designed as a specific selectable marker of Gwonje-early. As shown in Figure 3, two of five sets of primers, GJ-SNP1 and GJSNP4, were effectively able to select the Gwonje-early mutant from the WT and other commercial citrus varieties, indicating the use of both the GJ-SNP1 and GJSNP4 primer sets as specific selection markers for Gwonje-early. The SNP sites of the GJ-SNP1 and GJ-SNP4 markers are located on the 5′ untranslated region of the glutamate receptor 3.2 gene and the intergenic region, respectively. This suggests that these two specific selection markers for Gwonje-early have no relation to the growth and development of the mutant fruits and only function as varietal selection markers. We also recently identified a specific selection marker using AS-PCR for another citrus mutant with a late-ripening fruit phenotype, Ara-unshiu, induced by gamma irradiation [17]. In Upland cotton (*Gossypium hirsutum*), selection markers for resistant and susceptible alleles of *Fusarium oxysporum* f. sp. *vasinfectum*, a destructive soil-borne fungal pathogen, were identified using AS-PCR [44]. Using the genomic sequences of the potato StCWIN1 gene obtained from 155 accessions, a kompetitive allele-specific PCR (KASP) marker, allele-T, was developed to distinguish allelic variation in potato genotypes for tuber starch content and dry matter [45]. In wheat (*Triticum aestivum*), eight functional KASP markers were identified to select 40 wheat genotypes for drought tolerance [46]. Mukri et al. (2023) reported a novel KASP marker used to differentiate kernel row numbers in tropical field corn [47]. AS-PCR-based SNP markers based on primer combinations can be useful for rapid and convenient marker-assisted selection associated with the functional genes of crop species.

In conclusion, we developed a new *C. unshiu* mutant using gamma irradiation, Gwonje-early, which has a phenotype consisting of an ovate shape, a protrusion of the upper part of the fruit, and a large fruit size. Several morphological traits and the sugar/acid contents of the Gwonje-early mutant were also compared with its WT control fruit. Using SNPs generated in Gwonje-early compared with those in WT plants, we also identified two specific selectable markers (GJ-SNP1 and GJ-SNP4) for Gwonje-early to distinguish it from other commercial citrus varieties using AS-PCR. 

## 4. Materials and Methods

### 4.1. Plant Materials

The citrus mutant Gwonje-early was isolated via the gamma irradiation of *C. unshiu* Marc. cv. Miyagawa-wase, which was also used as the WT. Both the Gwonje-early and WT plants were planted at the Research Institute for Subtropical Agriculture and Biotechnology of Jeju National University, Seogwipo, Republic of Korea. Young leaves were harvested and immediately frozen in liquid nitrogen for genome re-sequencing analysis. Mature fruits (late November fruits) of both Gwonje-early and WT plants were harvested for analysis of Gwonje-early fruit traits. Other citrus varieties used in this study included the following: Ara-unshiu, Jedae-unshiu, Chung-Gyeon (CG, *C. kiyomi*), Gam-Pyong (GP, *C. hybrrid* cv. Kanpei), Hanla-Bong (HB, *C. reticulata* Shiranui), and Kara-Hyang (KH, *C. hybrid* Natsumi).

### 4.2. Analysis of Fruit Traits

The Gwonje-early mutant and WT fruit traits were investigated according to the method of Eun and Kim (2022) [16]. Fruit weight was measured using an electronic indicator scale (CAS Co., Ltd., Yangju, Republic of Korea). The vertical diameter, transverse diameter, and peel thickness of the fruit were measured using a digital caliper (MITUTOYO Corporation, Kawasaki, Japan). Fruit hardness was measured using a fruit hardness meter (LUTRON FR-5105, Antala Staška, Czech Republic). The sugar content and acidity were measured in 4–5 mL of fruit juice according to the NH-2000 (HORIBA, Kyoto, Japan) instruction manual. Changes in the fruit peel color were measured using a chromometer (CR-400, MINOLTA, Tokyo, Japan). Both Gwonje-early and WT fruits were investigated using 10 fruits per tree each year (2022 and 2023). All statistical analyses were carried out using the IBM SPSS software (SPSS for Windows, version 20, SPSS Inc., Armonk, NY, USA). Significant differences between the samples were calculated using a *t*-test.

### 4.3. Whole-Genome Re-Sequencing, Detection of Single-Nucleotide Polymorphism (SNP) and Insertion/Deletion (InDel) Variants, and Gene Ontology (GO) Analysis

Genomic DNA from young leaves of Gwonje-early and WT plants was extracted using a cetyltrimethylammonium bromide method [48] and sequenced using the Illumina HiSeq X Ten platform from Macrogen Co. (Daejeon, Republic of Korea). The sequence pre-processing, alignment to the reference genome, SNP/InDel detection and classification, and GO analysis were performed as described by Eun and Kim (2021) [26]. The reference genome information was obtained from the National Center for Biotechnology Information (NCBI) database (*C. unshiu* Marc. Miyagawa-wase (assembly CUMW_v1.0)) and the GenBank assembly (accession: GCA_002897195.1). Original sequence data from this study can be found in the NCBI Sequence Read Archive data libraries under the following accession numbers: PRJNA745525 (Miyagawa-1, WT) and PRJNA1048115 (Gwonje-early).

### 4.4. Allele-Specific PCR (AS-PCR)

The AS-PCR method was used to identify specific selection markers for the Gwonje-early mutant. The AS-PCR primers were designed according to Heo et al. (2023) [17]. Five homo-type SNPs were selected from the SNP variants between Gwonje-early and the WT control. For the allele-specific primer with the corresponding target SNP of Gwonje-early, an additional nucleotide mismatch was introduced at the second- or third-base position upstream of the 3′ terminus. The other primer employed the same sequence for both Gwonje-early and the WT. For the PCR amplification control, the same forward and reverse primers were used to amplify the PCR products of both Gwonje-early and the WT. PCR was performed in a final volume of 20 µL using TOPsimple DryMIX-nTaq (Enzynomics, Seoul, Republic of Korea), primers (5 µM), and 50 ng of extracted DNA. The PCR cycling conditions were as follows: initial denaturing at 95 °C for 3 min; 30 cycles of 95 °C for 30 s, 55 °C for 30 s, and 72 °C for 20 s; and a final extension at 72 °C for 5 min. The PCR products were electrophoresed on 1.5% (*w*/*v*) agarose gels. In addition to DNA samples from the Gwonje-early mutant and WT, DNA samples from other commercial citrus varieties and mutants were also examined. 

## Figures and Tables

**Figure 1 plants-13-00911-f001:**
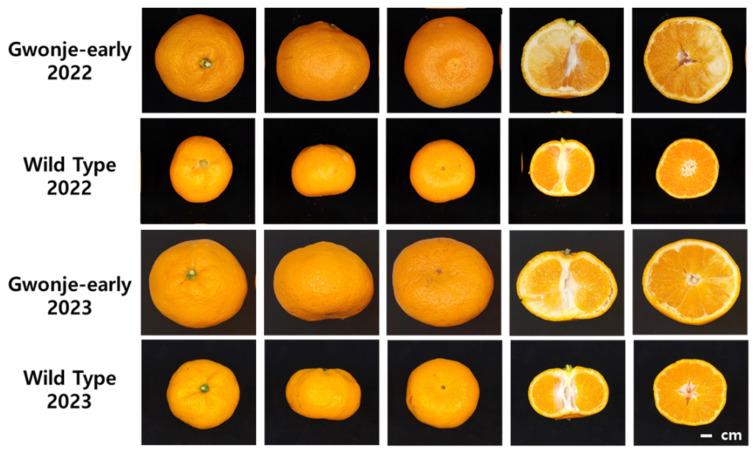
Morphological comparison between Gwonje-early and wild-type (WT) fruits.

**Figure 2 plants-13-00911-f002:**
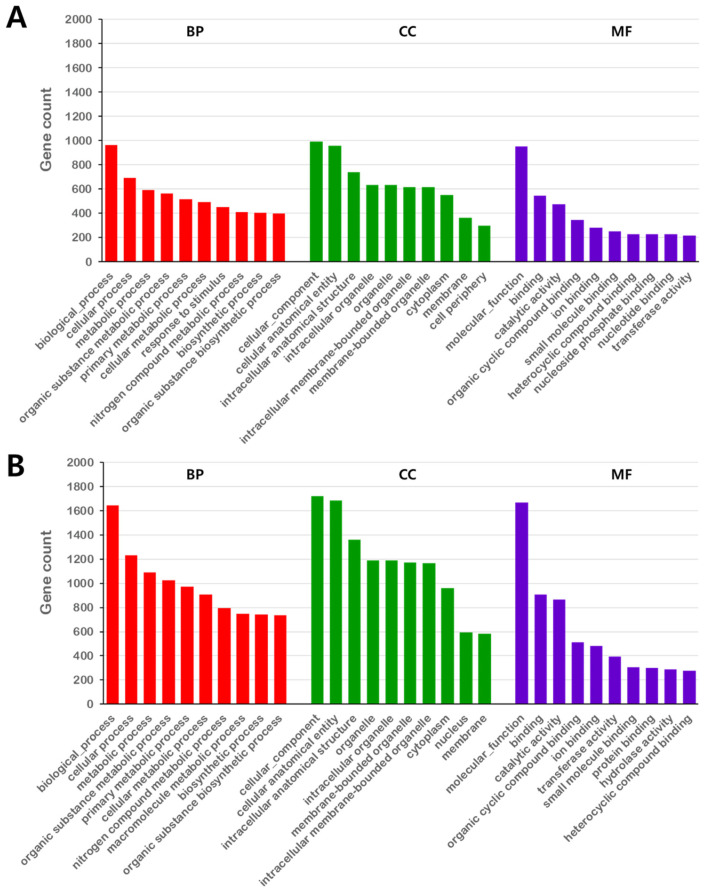
GO functional enrichment of InDel and SNP variants in Gwonje-early. (**A**) Top 10 biological process (BP), cellular component (CC), and molecular function (MF) InDel variants. (**B**) Top 10 BP, CC, and MF SNP variants.

**Figure 3 plants-13-00911-f003:**
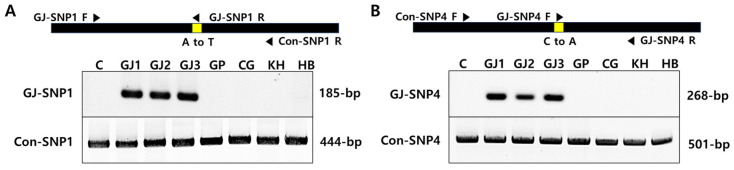
Identification of the Gwonje-early selection markers GJ-SNP1 (**A**) and GJ-SNP4 (**B**) by allele-specific PCR (AS-PCR). C: WT control; GJ1 to GJ3: Gwonje-early; GP: Gam-Pyong; CG: Chung-Gyeon; KH: Kara-Hyang; HB: Hanla-Bong.

**Table 1 plants-13-00911-t001:** Comparative analysis of wild type (WT) control and Gwonje-early fruits.

Year		2022			2023		
		WT	Gwonje-Early	*p*	WT	Gwonje-Early	*p*
Horizontal length (mm)		47.83 ± 3.68	68.24 ± 3.81	**	48.63 ± 2.79	65.70 ± 3.66	**
Vertical length (mm)		62.32 ± 6.64	78.63 ± 4.55	**	64.81 ± 5.04	84.93 ± 4.68	**
Single Fruit Weight (g)		98.94 ± 29.29	195.7 ± 21.60	**	111.44 ± 15.98	217.60 ± 21.40	**
Peel thickness (mm)		2.65 ± 0.43	3.06 ± 0.64	NS	2.59 ± 0.95	3.60 ± 0.43	*
Hardness (G)		862 ± 119	1071 ± 159	*	870 ± 259	1196 ± 476	*
Hunter color values	L (Light)	59.47 ± 1.62	58.26 ± 1.72	NS	58.48 ± 1.90	59.84 ± 2.56	NS
	a (Red)	25.92 ± 2.01	24.20 ± 2.20	NS	22.81 ± 2.15	25.72 ± 1.71	NS
	b (Yellow)	35.28 ± 1.05	34.02 ± 1.53	NS	35.75 ± 1.00	35.95 ± 1.56	NS
Total soluble solid (Brix)		9.41 ± 0.32	9.10 ± 0.30	NS	8.65 ± 1.05	8.90 ± 0.20	NS
Acidity (wt %)		0.46 ± 0.03	0.70 ± 0.03	**	0.51 ± 0.07	0.68 ± 0.02	**

* *p* < 0.05; ** *p* < 0.001; NS, not significant.

**Table 2 plants-13-00911-t002:** Re-sequencing data for WT and Gwonje-early plants.

Sample	Clean Reads ^1^	Mapped Reads ^2^	Mapped Regions ^3^	Coverage ^4^
WT	80,693,250	71,296,288 (88.35%)	310,983,030 (86.75%)	27.49×
Gwonje-early	96,356,847	87,217,702 (90.52%)	312,469,061 (86.88%)	29.97×

^1^ The number of clean reads that passed pre-processing and were used for read alignment. ^2^ The number of clean reads that mapped to the reference genome sequence when aligned. ^3^ The total length (in base pairs) of aligned reads. The percentage is in regard to the reference genome. ^4^ The value obtained by dividing the total read length of each sample by the assembled genome size (359.7 Mb).

**Table 3 plants-13-00911-t003:** Quantification of single-nucleotide polymorphisms (SNPs) and insertions/deletions (InDels) in only Gwonje-early compared with WT.

		Number ^1^	Homozygous	Heterozygous
Gwonje-early	SNP	6132	37	6096
	InDel	5969	29	5940

^1^ SNP and InDel loci showing differences in nucleotide sequences between comparative samples.

**Table 4 plants-13-00911-t004:** Genes containing SNPs and InDels in only Gwonje-early compared with WT.

		Number	GO Genes ^1^
Gwonje-early	SNP	3344	465
	InDel	3154	709

^1^ Number of genes containing SNPs or InDels and having enriched Gene Ontology (GO) annotations (e-value ≤ 1 × 10^−10^, best hits).

## Data Availability

Original sequence data can be found in the NCBI Sequence Read Archive under the following accession numbers: PRJNA745525 for the WT (Miyagawa-1) and PRJNA1048115 for Gwonje-early.

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
