# Peer review of "Characterization of a New Citrus Mutant Induced by Gamma Irradiation with a Unique Fruit Shape, Gwonje-Early, and Determination of Specific Selection Markers Using Allele-Specific PCR"

_plants, 2024, doi:10.3390/plants13060911_

Round 1

Reviewer 1 Report

Comments and Suggestions for Authors

Table 1 and 3 need statistical analysis

Please check figure 2 for a label (see file)

Author Response

Response to reviewer 1

First of all, we are grateful to reviewer 1 and editor for the valuable comments on this article and thank you for the time and consideration given in this regard. The response to reviewer 1 is as follows:

Comments and Suggestions for Authors

#1 Table 1 and 3 need statistical analysis.

-> We added the statistical analysis and combined the table 1 and table 2. Also we updated the results for both 2022 and 2023

#2 Please check Figure 2 for a label (see file)

-> We changed the label of figure 2

Reviewer 2 Report

Comments and Suggestions for Authors

In this study, Eun et al., reported to character an improved citrus of Gwonje-early, including fruit shape, color and contents, by using gamma irradiation. Also, they performed the re-sequencing and analysis of SNP and InDel markers between reference, WT and mutant. Two homo-type SNPs snd a InDel were annotated in the glutamate receptor 3.2, the hypothetical protein CUMW_259270, and NO-associated protein 1. Furthermore, they develop a AS-PCR markers based on A to T substitution on glutamate receptor 3.2 to identify Gwonje-early. This is a clearly written manuscript with some interesting results. However, the quality of some figures and tables, and the details in methods and discussion, needs to be improved.

Major concerns:

1. The details for developing GeSNP, does it derive from SNPs in the glutamate receptor 3.2? This is not found in methods or results section.

2. Figure 1, the WT for both 2021 and 2022 should listed in parallel. Also, please re-organize the order of the WT to maintain consistency.

3. In discussion section, Authors have talked a lot of genes involved in fruit growth and development, however, I have not caught the glutamate receptor 3.2, do this gene associate with any characteristic traits with Gwonje-early. If not, the negative citrus variety/variations in Figure 3 are quite limited.

4. Table 1 and 2, Table 4 and 5 could be integrated to one table for each two.

5. Table 2, Pay attention on the double lines.

6. Table 3, the symbol ≒ should be removed, or annotated the mean.

Comments on the Quality of English Language

English language is fine. 

Author Response

First of all, we are grateful to reviewer 1 and editor for the valuable comments on this article and thank you for the time and consideration given in this regard. The response to reviewer 2 is as follows:

Major concerns:

#1. The details for developing GeSNP, does it derive from SNPs in the glutamate receptor 3.2? This is not found in methods or results section.

We added in the result section as follows:

“Of the selected five GJ-SNPs, GJ-SNP1 was in the 5’-UTR of glutamate receptor 3.2 gene and the others were in intergenic regions.”

#2. Figure 1, the WT for both 2021 and 2022 should listed in parallel. Also, please re-organize the order of the WT to maintain consistency.

We changed the figure 1 as comments and also we update the results for both 2022 and 2023

#3.  In discussion section, Authors have talked a lot of genes involved in fruit growth and development, however, I have not caught the glutamate receptor 3.2, do this gene associate with any characteristic traits with Gwonje-early. If not, the negative citrus variety/variations in Figure 3 are quite limited.

We added in the discussion section as follows:

“The SNP sites of the GJ-SNP1 and GJ-SNP4 markers are located on the 5’ untranslated region of the glutamate receptor 3.2 gene and intergenic region, respectively. It suggested that these two specific selection markers for Gwonje-early have no relation to the growth and development of the mutant fruits and only function as varietal selection markers.”

#4. Table 1 and 2, Table 4 and 5 could be integrated to one table for each two.

We combined the table1 and table2 and also added the statistical analysis. And the table 4 and 5 (new table 3 and 4) we hope to be separated to understand more easily. 

  1. Table 2, Pay attention on the double lines.

We changed the table 2

  1. Table 3, the symbol “≒” should be removed, or annotated the mean.

We removed the symbol “≒” 

Reviewer 3 Report

Comments and Suggestions for Authors

This article reported a gamma irradiation induced citrus mutant with fruit shape change. Specific selection markers (Ge-SNP1 and Ge-SNP4) were also identified for this new fruit shape-related citrus mutant Gwonje-early. The results of this work provide valuable new germplasm and DNA markers for molecular breeding of citrus species, especially for citrus breeding focuses on fruit shapes. Following are some minor suggestions.

1. For Figure 1, suggest the authors exchange the order of the transverse and vertical-cut fruit picture of wild-type, which will be consistent with those of Gwonje-early. Besides, scale bars can also be added to these fruit pictures.

2. To accurately compare the differences between WT and Gwonje-early, statistic analysis could be added in Table 1 and Table 2.

3. For Table 3, the ‘%’ in the table header ‘Mapped regions (%)’ can be removed.

4. For the format of tables, please correctly use the three-line tables.

5. GeSNP1 … GeSNP5 or Ge-SNP1 … Ge-SNP5, please keep consitant within the whole manuscript. Similarly, Gwonje-early or Gwonje early.

Author Response

First of all, we are grateful to reviewer 1 and editor for the valuable comments on this article and thank you for the time and consideration given in this regard. The response to reviewer 3 is as follows:

#1. For Figure 1, suggest the authors exchange the order of the transverse and vertical-cut fruit picture of wild-type, which will be consistent with those of Gwonje-early. Besides, scale bars can also be added to these fruit pictures.

We changed the figure 1 as comments and also we update the results for both 2022 and 2023

#2. To accurately compare the differences between WT and Gwonje-early, statistic analysis could be added in Table 1 and Table 2.

We added the statistical analysis and combined the table 1 and table 2. Also we updated the results for both 2022 and 2023

#3. For Table 3, the ‘%’ in the table header ‘Mapped regions (%)’ can be removed.

We removed the “%”

  1. For the format of tables, please correctly use the three-line tables.

 We correctly changed the table format

  1. GeSNP1 … GeSNP5 or Ge-SNP1 … Ge-SNP5, please keep consitant within the whole manuscript. Similarly, Gwonje-earlyor Gwonjeearly.

We consistently changed the names

Reviewer 4 Report

Comments and Suggestions for Authors

The definition of "mutation" that you present in the 1st sentence of the second paragraph of the Introduction is a bit odd. I checked your citation. If you stick to this definition, you should put it in quotations since it is quoted verbatim from the source. I think the Oxford English Dictionary has a clearer definition that fits better with the kind of things you describe in the study. According to the Oxford dictionary, mutation is "the changing of the structure of a gene, resulting in a variant form that may be transmitted to subsequent generations, caused by the alteration of single base units in DNA, or the deletion, insertion, or rearrangement of larger sections of genes or chromosomes." Or you might do better to put it in your own words, rather than quote the unusual definition you chose in this draft.

Other than that, the text reads well and you explained your study, purpose, methods and results adequately and concisely. Your discussion is also good. Tables 4 and 5 might be a bit clearer if you place two lines in the first column identifying your two rows as 'WT' on the first row and 'Gwonje-early' on the second row as you did in the previous tables, rather than saying "WT vs Gwonje-early".

Similarly, in the legend of Figure 2, you should clearly state that the bar graphs in part A are for WT and those for part B are for Gwonje-early.

The last two labels atop your gel images in Figure 3 (in both parts A and B) should be changed from 'KR' to 'KH' and 'HL' to 'HB' to match your legend for that figure and the mention of these two cultivars in your text. 

Other than these small things, I found your paper easy to follow and interesting. It's nice to see people still having success with mutation breeding. You do well to point out its advantage over other techniques. It is also great that you did a genome-wide comparison of your starting material and mutant product and actually found markers to distinguish them.

Well done!

Author Response

First of all, we are grateful to reviewer 1 and editor for the valuable comments on this article and thank you for the time and consideration given in this regard. The response to reviewer 4 is as follows:

#1 Tables 4 and 5 might be a bit clearer if you place two lines in the first column identifying your two rows as 'WT' on the first row and 'Gwonje-early' on the second row as you did in the previous tables, rather than saying "WT vs Gwonje-early".

We changed the table 3 and 5 (new table 3 and 4) to be understand easily.

Similarly, in the legend of Figure 2, you should clearly state that the bar graphs in part A are for WT and those for part B are for Gwonje-early.

In the figure 2, the part A is for GO result of InDel variants and part B is for SNP variants in Gwonje-early.

The last two labels atop your gel images in Figure 3 (in both parts A and B) should be changed from 'KR' to 'KH' and 'HL' to 'HB' to match your legend for that figure and the mention of these two cultivars in your text. 

We correctly changed the names.

Reviewer 5 Report

Comments and Suggestions for Authors

The work of Chang-Ho Eun et al. addresses an interesting topic for quality improvement in the citrus industry worldwide. The induction of mutations by gamma rays has been (and is currently) used for the production of new genotypes. Likewise, the detection of induced variations in the genome and the development of specific molecular markers is also the subject of work in many citrus breeding programs. The work is well described and executed, but I find the main limitation in that it is very specific work without revealing or describing methodologies sufficiently innovative to be considered as a scientific novelty. Therefore, although the work is well focused, I would understand that it is justified to publish it in a journal with a different scope. 

Comments on the Quality of English Language

The quality of the language is adequate.

Author Response

Response to reviewer 5

First of all, thank you for the time and consideration given in this regard. We received the 5 reviewer’s comments for this study. And the 4 of 5 reviewers reviewed our study to be edited somewhere with mistake or misspelled or to be added statistical analysis. So, we hope to be published in this time if you don’t mind.

Best regards

Round 2

Reviewer 2 Report

Comments and Suggestions for Authors

The author had addressed all my concerns. The modification is fine. I have no any other comments. Suggest to accept.